

# A flexible and accurate method for electroencephalography rhythms extraction based on circulant singular spectrum analysis

Hai Hu, Zihang Pu and Peng Wang

Department of Precision Instrument, Tsinghua University, Beijing, China

## ABSTRACT

Rhythms extraction from electroencephalography (EEG) signals can be used to monitor the physiological and pathological states of the brain and has attracted much attention in recent studies. A flexible and accurate method for EEG rhythms extraction was proposed by incorporating a novel circulant singular spectrum analysis (CiSSA). The EEG signals are decomposed into the sum of a set of orthogonal reconstructed components (RCs) at known frequencies. The frequency bandwidth of each RC is limited to a particular brain rhythm band, with no frequency mixing between different RCs. The RCs are then grouped flexibly to extract the desired EEG rhythms based on the known frequencies. The extracted brain rhythms are accurate and no mixed components of other rhythms or artifacts are included. Simulated EEG data based on the Markov Process Amplitude EEG model and experimental EEG data in the eyes-open and eyes-closed states were used to verify the CiSSA-based method. The results showed that the CiSSA-based method is flexible in alpha rhythms extraction and has a higher accuracy in distinguishing between the eyes-open and eyes-closed states, compared with the basic SSA method, the wavelet decomposition method, and the finite impulse response filtering method.

## INTRODUCTION

Electroencephalograms (EEGs) are the electrical activity of the brain's neurons recorded at the scalp surface (*Henry, 2006*). They consist of several rhythm bands: delta (1–4 Hz), theta (4–8 Hz), alpha (8–13 Hz), beta (13–30 Hz) and gamma (>30 Hz). Because the rhythms reflect different physiological and pathological information, EEG rhythms extraction has been widely applied in many areas. Examples include portable and wearable EEG devices (*Hwang et al., 2018*; *Maskeliunas et al., 2016*), mental fatigue assessment (*Taran & Bajaj, 2017*), disease diagnosis (*Babiloni et al., 2016*; *Gupta & Pachori, 2019*), and brain computer interface systems (*Jeunet et al., 2019*; *Liu et al., 2020*).

The accuracy of EEG rhythms extraction determines the physiological and pathological information it provides. Various methods have been proposed to extract the desired EEG

Corresponding author
Peng Wang,
peng@mail.tsinghua.edu.cn

rhythms. Filtering components have the ability to restrict a signal to a specific frequency band, and such bandpass filters were first used to extract EEG rhythms (*Pfurtscheller et al., 1997*). This method performed well in EEGs of high signal-to-noise ratio (SNR). Then, the wavelet transform (WT) method was used for EEG rhythms extraction (*Duque-Muñoz, Pinzon-Morales & Castellanos-Dominguez, 2015*). By estimating the rhythms with a customized wavelet, the WT method can extract time-varying EEG rhythms with changes in brain state. To facilitate the EEG rhythms extraction, the independent component analysis (ICA) method was then introduced (*Kavuri, Veluvolu & Chai, 2018*). By incorporating priori information about the desired rhythms as reference signals, the ICA method can extract EEG rhythms automatically. However, the extracted rhythms using the bandpass filter, WT, and ICA methods were contaminated by noise and artifacts overlapping in time–frequency space. In recent years, to improve the accuracy of EEG rhythms extraction, the singular spectrum analysis (SSA) method has been used (*Akar et al., 2015*; *Mohammadi et al., 2016*). This nonparametric method enables the separation of different sources even when they overlap in time–frequency space (*Mohammadi et al., 2015*).

The SSA method is a nonparametric signal processing method proposed to extract useful information from experimental data (*Broomhead & King, 1986*). SSA consists of two stages: decomposition and reconstruction. Decomposition involves time-delay embedding called Takens' theorem (*Takens, 1981*), followed by singular value decomposition (SVD) (*Mees, Rapp & Jennings, 1987*). Reconstruction involves grouping and diagonal averaging (*Vautard, Yiou & Ghil, 1992*). SSA works well for single channel signals as well as multichannel signals. It allows the exploitation of SSA for a variety of applications including biomedical signal processing such as signal restoration, change detection, segmentation, anomaly detection and prediction (*Sanei & Hassani, 2015*; *Xu et al., 2018*). Compared with the narrowband filter methods which are not able to separate the component mixing overlapping in frequency space, the SSA method is utilized to find the structure of nonlinear and non-stationary signal and enables the separation of different sources even overlapping in the time-frequency space. Therefore, SSA has been successfully applied in the research of artifacts removal and rhythms extraction from the EEG signal (*Maddirala & Shaik, 2016a*, *2016b*; *Mohammadi et al., 2015*; *Xu et al., 2018*).

In the basic SSA method, the grouping rule is important for SSA reconstruction. However, because of the lack of the information about the amplitude and frequency of the reconstructed components (RCs), there is no general grouping rule. Different grouping rules have been proposed depending on the research target, the types of signals, and noise. The conventional SSA grouping is performed according to the magnitudes of eigenvalues related to the power of each RC (*De Carvalho & Rua, 2017*; *Teixeira et al., 2005*; *Yang et al., 2016*). *Mohammadi et al. (2016)* proposed a new grouping rule based on eigenvalue pairs to extract the main rhythms from sleep EEG signals. *Hai et al. (2017)* proposed another efficient grouping rule based on the similarity between the eigenvalues and the peak frequency of RC, which makes SSA adaptive to EEG signals containing different levels of artifacts and rhythms. However, these grouping rules can only be applied to specific types of signals and must be incorporated with other methods (*e.g.*, Fourier transform or wavelet decomposition) to pre-identify the frequencies of RCs, which is time-consuming

and inflexible. Besides, the observed frequency mixing between different RCs leads to inaccurate EEG rhythms extraction (*Xu et al., 2018*).

In this paper, we introduce a novel circulant singular spectrum analysis (CiSSA) method (*Bógalo, Poncela & Senra, 2021*) on the basis of SSA to improve the flexibility and accuracy of EEG rhythms extraction. Compared with the basic SSA method, the CiSSA method has the advantage of avoiding the need for pre-identifying the frequencies of RCs. A set of orthogonal vectors are obtained by decomposing the circulant matrix, and the EEG signals can be decomposed into the sum of a set of orthogonal RCs of known frequencies. The RCs can be grouped automatically and flexibly to extract the specific EEG rhythms based on their frequencies. In addition, because the frequency bandwidth of each RC is limited to a particular band of the brain rhythm of interest and the artifacts are removed by the basic SSA, the extracted brain rhythms are accurate and no mixed components of other rhythms and artifacts are included.

## METHODS

The CiSSA method is a nonparametric signal extraction method proposed by *Bógalo, Poncela & Senra (2021)*. CiSSA consists of four steps: embedding, decomposition, diagonal averaging, and grouping. As in the basic SSA method, in the time-delay embedding step, the single-channel EEG time series $\mathbf{s} = (s_1, s_2, \ldots, s_N)^T$ (superscript $T$ denotes the transpose of a vector) is mapped onto a multidimensional trajectory matrix $\mathbf{X}$ using a sliding window (*Takens, 1981*):

$$\mathbf{X} = (\mathbf{S}_1, \mathbf{S}_2, \ldots, \mathbf{S}_K) = \begin{pmatrix} s_1 & s_2 & \cdots & s_K \\ s_2 & s_3 & \cdots & s_{K+1} \\ \vdots & \vdots & \ddots & \vdots \\ s_L & s_{L+1} & \cdots & s_N \end{pmatrix} \tag{1}$$

where $L$ denotes the window length (or embedding dimension), $K = N - L + 1$, and $\mathbf{S}_i$ denotes the lagged vector.

In the decomposition step, the trajectory matrix is decomposed into elementary matrices of rank 1 that are associated with different frequencies. To do so, a related circulant matrix $\mathbf{C}_L$ is built based on the second order moments of the time series (*Bógalo, Poncela & Senra, 2021*):

$$\mathbf{C}_L(f) = \begin{pmatrix} c_0 & c_1 & c_2 & \cdots & c_{L-1} \\ c_{L-1} & c_0 & c_1 & \cdots & c_{L-2} \\ \vdots & \vdots & \vdots & \ddots & \vdots \\ c_1 & c_2 & c_3 & \cdots & c_0 \end{pmatrix} \tag{2}$$

where

$$c_m = \frac{L-m}{L}\gamma_m + \frac{m}{L}\gamma_{L-m}, \gamma_m = \frac{1}{N-m}\sum_{t=1}^{T-m} s_t s_{t+m}, \quad m = 0, 1, \ldots, L-1 \tag{3}$$

The eigenvalues and eigenvectors of $\mathbf{C}_L$, respectively, are given (*Gray, 2006*)
$$\lambda_k = \sum_{m=0}^{L-1} c_m \exp\left(i2\pi m \frac{k-1}{L}\right) = f\left(\frac{k-1}{L}\right), k = 1, 2, \dots, L \tag{4}$$

$$\mathbf{u}_k = L^{-1/2}\left(u_{k,1}, u_{k,2}, \dots, u_{k,L}\right)^H, u_{k,j} = \exp\left(-i2\pi(j-1)\frac{k-1}{L}\right), k = 1, 2, \dots, L$$

where $f(\cdot)$ denotes the power spectral density of the signal, and $H$ indicates the conjugate transpose of a matrix. The $k$-th eigenvalue and the corresponding eigenvector are associated with the given frequencies by

$$f_k = \frac{k-1}{L} f_s \tag{5}$$

where $f_s$ is the sampling rate of the EEG signals. As a consequence, the diagonalization of $\mathbf{C}_L$ allows us to write $\mathbf{X}$ as the sum of the elementary matrices $\mathbf{X}_k$:

$$\mathbf{X} = \sum_{k=1}^{L} \mathbf{X}_k = \sum_{k=1}^{L} \mathbf{u}_k \mathbf{u}_k^H \mathbf{X} \tag{6}$$

The symmetry of the power spectral density leads to $\lambda_k = \lambda_{L+2-k}$. The corresponding eigenvectors given by Eq. (4) are complex; therefore, they are paired with complex conjugates, $\mathbf{u}_k = \mathbf{u}_{L+2-k}^*$ where $\mathbf{u}^*$ indicates the complex conjugate of a vector $\mathbf{u}$. Then, $\mathbf{X}_k$ and $\mathbf{X}_{L+2-k}$ correspond to the same frequency $f_k$.

To obtain the elementary matrices by frequency, we first form the groups of two elements $B_k = \{k, L+2-k\}, k = 2, 3, \dots, M, M = \lfloor(L+1)/2\rfloor, \lfloor \cdot \rfloor$ is the integer part operator, with $B_1 = \{1\}$ and $B_{L/2+1} = \{L/2 + 1\}$ if $L$ is even. Then, the real elementary matrix $\mathbf{X}_{B_k}$ is computed as the sum of the two elementary matrices $\mathbf{X}_k$ and $\mathbf{X}_{L+2-k}$, which are associated with eigenvalues $\lambda_k$, $\lambda_{L+2-k}$ and frequency $f_k$, given by Eq. (5)

$$\begin{aligned}\mathbf{X}_{B_k} &= \mathbf{X}_k + \mathbf{X}_{L+2-k} = \mathbf{u}_k \mathbf{u}_k^H \mathbf{X} + \mathbf{u}_{L+2-k} \mathbf{u}_{L+2-k}^H \mathbf{X} = (\mathbf{u}_k \mathbf{u}_k^H + \mathbf{u}_k^* \mathbf{u}_k')\mathbf{X} \\ &= 2(R_{\mathbf{u}_k} R_{\mathbf{u}_k}' + I_{\mathbf{u}_k} I_{\mathbf{u}_k}')\mathbf{X}\end{aligned} \tag{7}$$

where $R_{\mathbf{u}_k}$ and $I_{\mathbf{u}_k}$ denote the real and imaginary parts of $\mathbf{u}_k$, respectively, and the matrices $\mathbf{X}_{B_k}$ are real.

Then, in the diagonal averaging step (*Vautard, Yiou & Ghil, 1992*), several time series are reconstructed from the corresponding real elementary matrices $\mathbf{X}_{B_k}$. The reconstructed time series are generally called RCs. Theoretically, the frequencies of the RCs are given by Eq. (5). Finally, the alpha rhythm (8–13 Hz) can be extracted automatically by

$$V_{alpha} = \sum_{i=\lceil 1+8L/f_s \rceil}^{\lfloor 1+13L/f_s \rfloor} RC_i \tag{8}$$

The frequency bandwidth of each RC can be roughly expressed by *Bozzo, Carniel & Fasino (2010)* and *Xu et al. (2018)*

$$f_b = f_s/L \tag{9}$$

As a consequence, the frequency bandwidth of each RC is limited to $f_s/L$. Considering the frequency of each RC given by Eq. (5), there is no frequency mixing between

| Algorithm 1 The pseudo-code of the CiSSA method for alpha rhythm extraction. |
| --- |
| **input s, L**: single-channel EEG time series **s** and embedding dimension $L$ |
| **output α**: extracted alpha rhythm |
| **procedures** |
| (1) **X**: the trajectory matrix is constructed by Eq. (1) |
| (2) **C**$_L$: the circulant matrix is built by Eqs. (2) and (3) |
| (3) $\lambda_k$, **u**$_k$: the circulant matrix **C**$_L$ is decomposed, and a set of eigenvalues and eigenvectors is derived by Eq. (4) |
| (4) **X**$_{B_k}$: the real elementary matrices are derived by Eq. (7) |
| (5) **α**: the RCs are grouped by Eq. (8) to obtain the alpha rhythm |
| **return α** |

different RCs, and the extracted alpha rhythms do not contain mixed components of other rhythms. In order to remove the frequency components outside $f_{band}$, with $f_{band}$ denoting the bandwidth of the brain rhythm of interest (alpha rhythm), the frequency bandwidth $f_b$ is less than the bandwidth of the brain rhythm of interest $f_{band}$ ($f_b \leq f_{band}$). Therefore the embedding dimension $L$ should satisfy the condition $L \geq f_s/f_{band}$.

The pseudo-code of the CiSSA method is shown in Algorithm 1. The Institutional Review Board of Tsinghua University granted ethical approval (20170010) to carry out the experiment within its facilities.

# SIMULATION RESULTS AND DISCUSSION

## Markov process amplitude EEG model

Simulated spontaneous EEG signals were used to verify the validity of the CiSSA method in alpha rhythm extraction. Spontaneous EEG signals were generated based on the Markov Process Amplitude (MPA) EEG model (*Bai et al., 2001*; *Nishida, Nakamura & Shibasaki, 1986*). The MPA EEG model is a powerful and widely used method to simulate and interpret EEG signals. With a few parameters, the model can represent the two major characteristics of EEG signals: rhythmic oscillation and randomness. Rhythmic oscillation is represented by sinusoidal waves, and randomness was represented by the stochastic process amplitude of the first-order Markov process. In recent years, the MPA EEG model has been applied in several studies analyzing spontaneous EEG signals, which have employed such techniques as feature expression, quantitative analysis (*Nakamura et al., 1997*), and algorithm verification (*Xu et al., 2018*).

In the MPA EEG model, EEG signals consist of several rhythmic oscillations expressed by a sinusoidal wave

$$s(n\Delta t) = \sum_{i=1}^{K} a_i(n\Delta t) \sin(2\pi f_i n\Delta t + \theta_i) \tag{10}$$

where $n$ is the number of samples, $\Delta t$ is the time interval, $K$ is the number of rhythms, $f$ is the dominant frequency of rhythm, $\theta$ is the initial phase (zero), and $a$

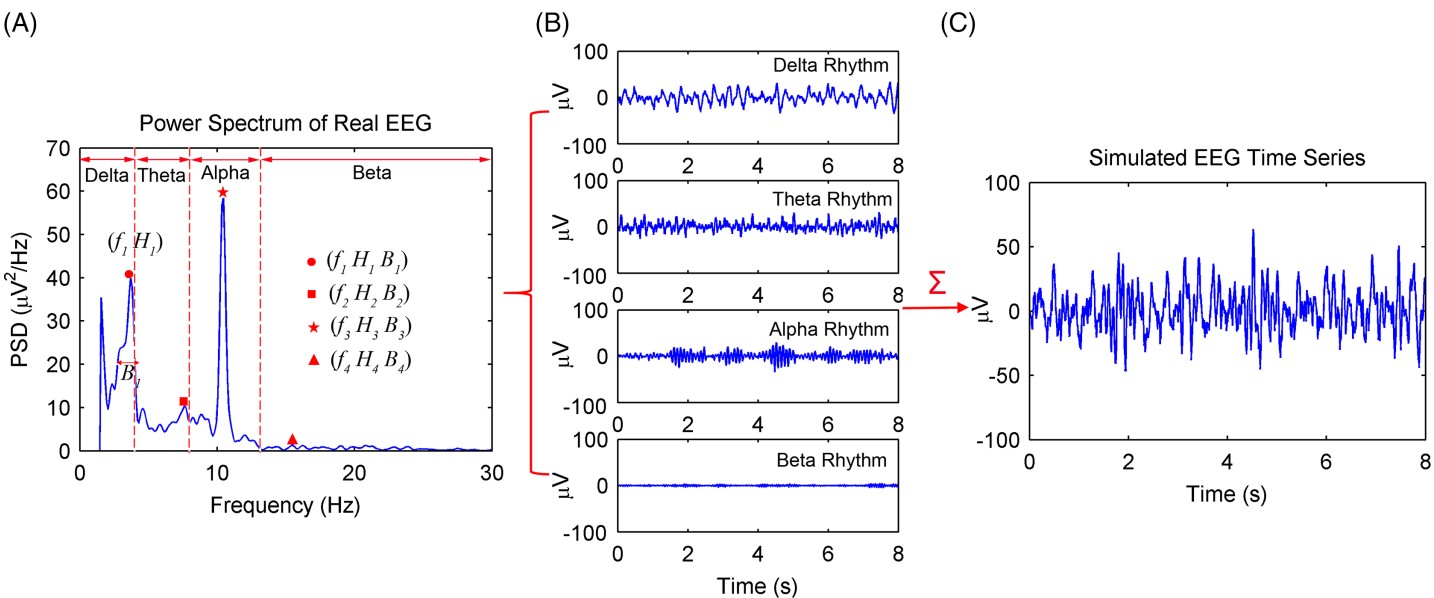

**Figure 1 Procedures of the spontaneous EEG simulation based on the MPA EEG model.** (A) The power spectrum of a real EEG. The peak frequencies ($f_i$), amplitude ($H_i$) and the frequency width ($B_i$) at half of amplitude of EEG rhythms were determined based on the power spectrum. (B) The simulated four rhythms: delta, theta, alpha and beta, based on the determined parameters. (C) The simulated spontaneous EEG generated by a combination of the four rhythms.                    

is the rhythmic amplitude obtained from the following first-order Gauss–Markov process:

$$a_i[(n+1)\Delta t] = \gamma_i a_i(n\Delta t) + \xi_i(n\Delta t) \qquad (11)$$

where $\gamma$ is the coefficient of the first-order Markov process, and $\xi$ is a random increment of Gaussian distribution with mean zero and variance $\sigma^\xi$. Therefore, the rhythmic amplitude at the succeeding time $(n+1)\Delta t$ depends only on the amplitude at time $\Delta t$ and is determined only by two parameters: $\gamma$ and $\sigma^\xi$. The parameters of the MPA EEG model are determined in the frequency domain to achieve the maximum likelihood with respect to the power spectrum of real EEG. $H_i$ is defined as the amplitude, and $B_i$ is the frequency width at half of $H_i$ of the EEG power spectrum. Based on the literature (*Bai et al., 2001*), $H_i$, $B_i$ can be described as

$$\begin{cases} H_i = \frac{\Delta t \left(\sigma_i^\xi\right)^2}{4(1-\gamma_i)^2} \\ B_i = \frac{1}{\pi \Delta t} \cos^{-1} \frac{4\gamma_i - 1 - (\gamma_i)^2}{2\gamma_i} \end{cases} \qquad (12)$$

The simulation procedures of spontaneous EEG signals based on the MPA EEG model are shown in Fig. 1. First, the power spectrum of a real EEG signal with sampling rate $f_s = 200$ Hz is calculated in Fig. 1A. Then, $f_i$, $H_i$, and $B_i$, which represent the peak frequencies, amplitude, and the frequency width at half of amplitude of the EEG rhythms (delta, theta, alpha, and beta), respectively, are obtained according to the power spectrum. Based on Eq. (12), the parameters of the first-order Gauss–Markov process ($\gamma$ and $\sigma^\xi$) are obtained. All parameters of the MPA EEG model are shown in Table 1. Then, the delta,

**Table 1 The parameters of the MPA EEG model.**

| Symbol | Value | Comments |
| --- | --- | --- |
| $f_1$ (Hz) | 3.71 | Delta rhythm |
| $\sigma_1^\xi$ | 3.53 | |
| $\gamma_1$ | 0.98 | |
| $f_2$ (Hz) | 7.62 | Theta rhythm |
| $\sigma_2^\xi$ | 4.35 | |
| $\gamma_2$ | 0.95 | |
| $f_3$ (Hz) | 10.45 | Alpha rhythm |
| $\sigma_3^\xi$ | 1.65 | |
| $\gamma_3$ | 0.99 | |
| $f_4$ (Hz) | 15.43 | Beta rhythm |
| $\sigma_4^\xi$ | 0.24 | |
| $\gamma_4$ | 0.99 | |

theta, alpha, and beta rhythms are simulated based on the determined parameters, as shown in Fig. 1B. Finally, the simulated EEG signal (shown in Fig. 1C) is generated as the sum of the four rhythms. The simulated spontaneous EEG lasts for 8 s with a 5-ms $\Delta t$ interval.

## Circulant singular spectrum analysis of simulated EEG signals

The simulated EEG signal is processed by the CiSSA method. The embedding dimension $L$ should satisfy the condition $L \geq f_s/f_{band} = 40$, with the data sampling rate $f_s = 200$ Hz and the bandwidth of alpha rhythm (8–13 Hz) $f_{band} = 5$ Hz. Moreover, the calculated frequency bandwidth $f_s/L$ is always less than the real frequency bandwidth of each reconstructed component. It is better to select the embedding length $L$ twice times as long as $f_s/f_{band}$. Therefore, the embedding length $L$ was set to 40 or 80.

Figure 2 shows the power spectrum density (PSD) of the first six reconstructed components (RCs) for a simulated EEG signal. When $L = 40$, as shown in Fig. 2A, every RC falls on the theoretical frequency derived by Eq. (5). Furthermore, the bandwidth of each RC is limited to $f_s/L = 5$ Hz. Similarly, when $L = 80$, as shown in Fig. 2B, all RCs fall on the theoretical frequencies with the bandwidth limited to 2.5 Hz. However, the simulated EEG signal is processed by the basic SSA method with the embedding dimension set to $L = 40$. The PSD of the first six RCs is shown in Fig. 3. The frequency of each RC is unknown. To group the RCs by frequency, other algorithms like Fourier transform are introduced to calculate the frequency of RCs. Besides, according to the PSD of RC5 and RC6, some components fall outside of the bandwidth limit of $f_s/L = 5$ Hz. This phenomenon is called component mixing.

Because the frequencies of the RCs processed by the CiSSA method are known, the alpha rhythm of the EEG signal can be extracted by Eq. (8). The error parameter for evaluating the performance of the alpha rhythm extraction is defined as *Xu et al. (2018)*
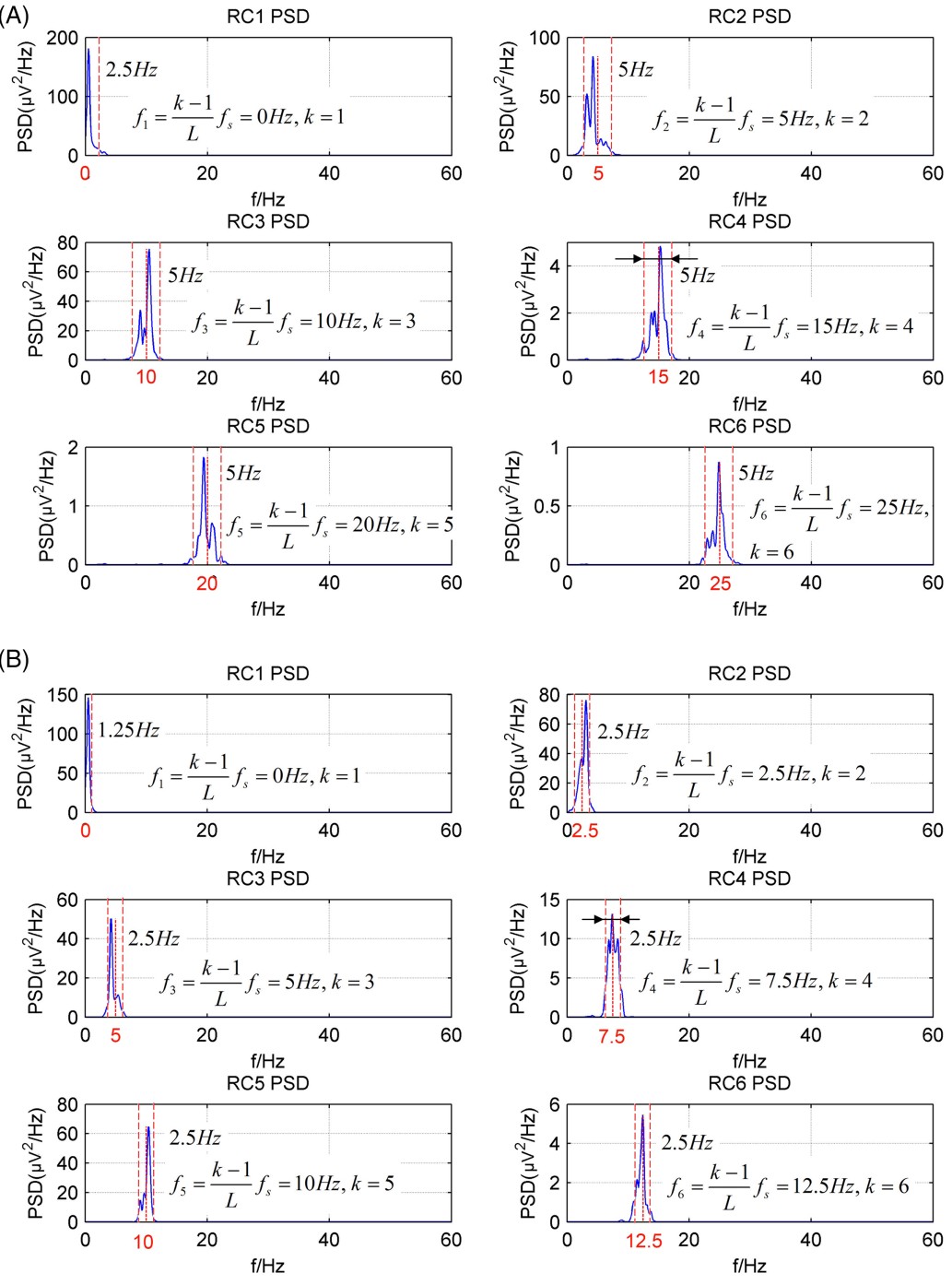

**Figure 2** **The power spectrum density of the first six reconstructed components of the simulated EEG signal processed by the CiSSA method.** (A) $L = 40$; (B) $L = 80$.

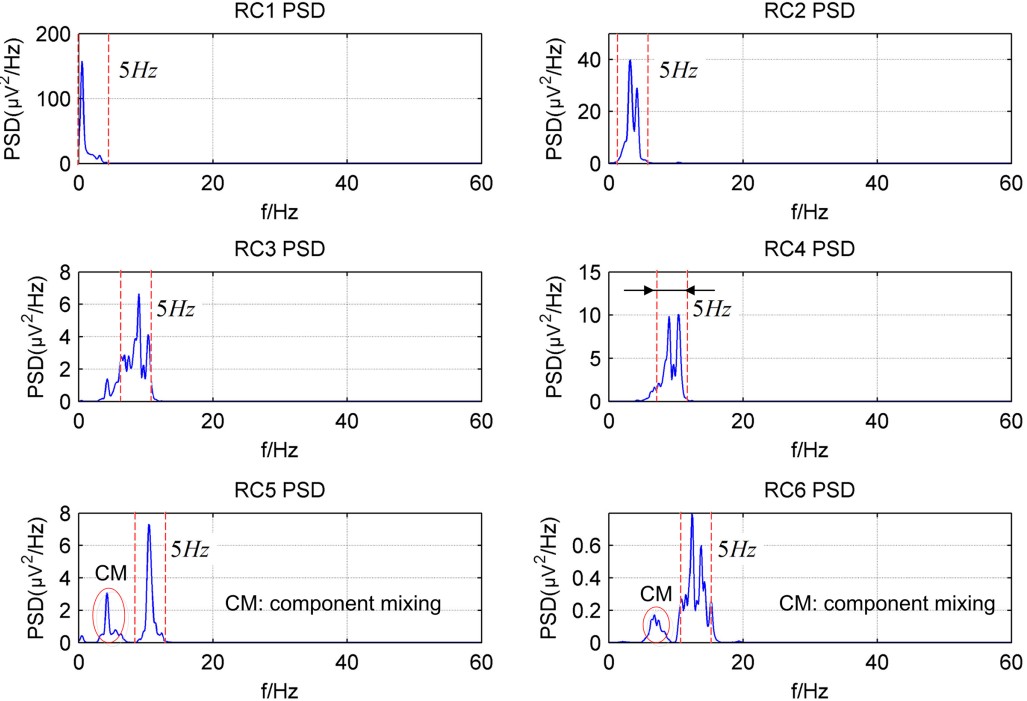

**Figure 3** The power spectrum density of the first six reconstructed components of the simulated EEG signal processed by the basic SSA method with the embedding dimension $L = 40$.

$$\varepsilon_{ave} = \frac{1}{N}\sum_{i=1}^{N}|P_\alpha(i) - P_e(i)| \tag{13}$$

where $\varepsilon_{ave}$ is the average error of the PSD between the simulated alpha rhythm and the extracted alpha rhythm, $P_\alpha(i)$ is the PSD of the simulated alpha rhythm, $P_e(i)$ is the PSD of the extracted alpha rhythm, and $N$ is the length of the PSD.

Figure 4A shows the extracted alpha rhythm of the simulated EEG signal by the CiSSA method with the embedding dimension set to $L = 40$. Figure 2A shows that RC3 represents the alpha rhythm. The PSD of the simulated and extracted alpha rhythm by the CiSSA method when $L = 40$ is shown in Fig. 4B. There was component mixing (slash shadow), and the error of the extracted alpha rhythm was $\varepsilon = 0.43 \ \mu V^2/\text{Hz}$. Similarly, RC5 and RC6 represent the alpha rhythm from Fig. 2B with an embedding dimension of $L = 80$. The alpha rhythm extracted from the combination of RC5 and RC6 is shown in Fig. 4C. The PSD of the simulated and extracted alpha rhythm by the CSSA method when $L = 80$ is shown in Fig. 4D. There was less component mixing (slashed shadow) than that observed when $L = 40$, and the error of the extracted alpha rhythm was $\varepsilon = 0.27 \ \mu V^2/\text{Hz}$. In the basic SSA method, the alpha rhythm was extracted according to the adaptive grouping rule (*Hai et al., 2017*). The extracted alpha rhythm by the basic SSA method when $L = 40$ is the sum of RC3, RC4, RC5, and RC6, as shown in Fig. 4E. The PSD of the

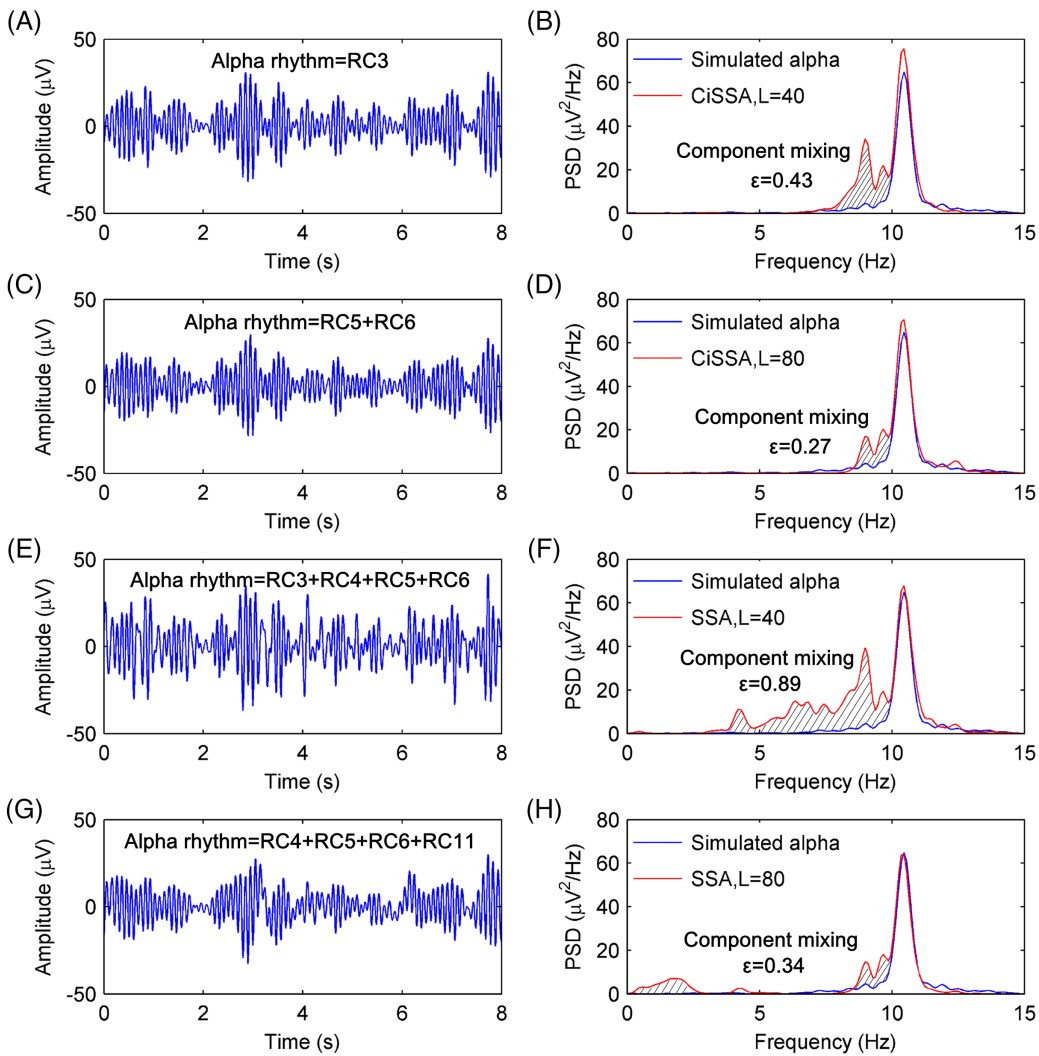

**Figure 4** **The extracted alpha rhythms of the simulated EEG signal and the PSD of the simulated and extracted alpha rhythms.** (A) The extracted alpha rhythm of the simulated EEG signal by the CiSSA method with the embedding dimension set to be $L = 40$. RC3 represents the alpha rhythm. (B) The PSD of the simulated and extracted alpha rhythm by CiSSA method when $L = 40$. The slash shadow part is the component mixing and the error of extracted alpha rhythms is 0.43 $uV^2$/Hz. (C) The extracted alpha rhythm of the simulated EEG signal by the CiSSA method with the embedding dimension set to be $L = 80$. RC3 and RC4 represent the alpha rhythm. (D) The PSD of the simulated and extracted alpha rhythm by CiSSA method when $L = 80$. The error of extracted alpha rhythms is 0.27 $uV^2$/Hz. (E) The extracted alpha rhythm of the simulated EEG signal by the basic SSA method with the embedding dimension set to be $L = 40$. RC3, RC4, RC5 and RC6 represent the alpha rhythm. (F) The PSD of the simulated and extracted alpha rhythm by the basic SSA method ($L = 40$). The error of extracted alpha rhythms is 0.89 $uV^2$/Hz. (G) The extracted alpha rhythm of the simulated EEG signal by the basic SSA method with the embedding dimension set to be $L = 80$. RC4, RC5, RC6 and RC11 represent the alpha rhythm. (H) The PSD of the simulated and extracted alpha rhythm by the basic SSA method ($L = 80$). The error of extracted alpha rhythms is 0.34 $uV^2$/Hz.  

simulated and extracted alpha rhythms by basic SSA when $L = 40$, shown in Fig. 4F, illustrates the presence of more component mixing (slashed shadow) than that found with the CiSSA method when $L = 40$, and the error of the extracted alpha rhythm was

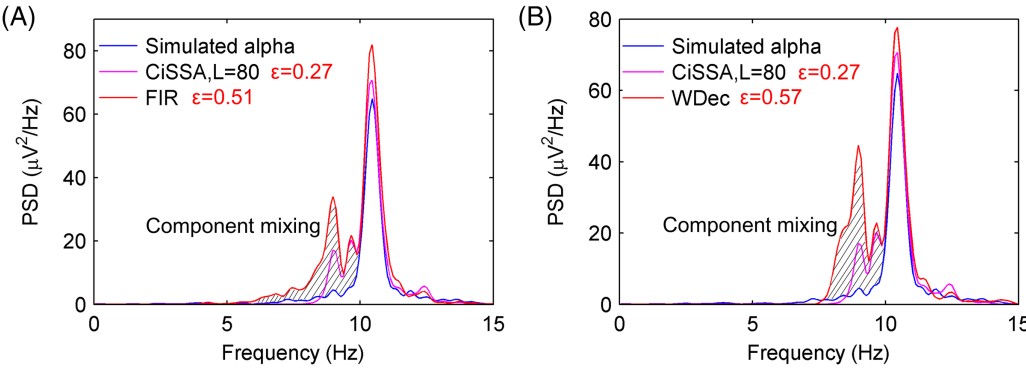

**Figure 5 The PSD of the extracted alpha rhythms by FIR and WDec method.** (A) The PSD of the simulated alpha rhythm and the extracted alpha rhythms by CiSSA and FIR. (B) The PSD of the simulated alpha rhythm and the extracted alpha rhythms by CiSSA and WDec.

$\varepsilon = 0.89\ \mu V^2/\text{Hz}$. Similarly, the extracted alpha rhythm by the basic SSA method when $L = 80$ is the sum of RC4, RC5, RC6, and RC11, as shown in Fig. 4G. Figure 4H shows the PSD of the simulated and extracted alpha rhythms by basic SSA when $L = 80$. The error of the extracted alpha rhythm by the basic SSA method ($L = 80$) was $\varepsilon = 0.34\ \mu V^2/\text{Hz}$, larger than that by the CiSSA method ($L = 80$). Besides, the extracted alpha rhythm by the basic SSA method ($L = 40$ or $L = 80$) contained the frequency components outside the alpha rhythm bands (8–13 Hz).

To compare the performance of the alpha rhythm extraction with that of other methods, the alpha rhythms were extracted by the finite impulse response (FIR) filtering methods and the wavelet decomposition (WDec) method. The FIR filter is a simple, stable and narrowband filter that has been utilized in many digital applications like image processing, wireless communication, biomedical etc. (*Khurshid & Mir, 2017*; *Mahabub, 2019*). The WDec can be performed by a series of high and low-pass filters (*Mamun, Al-Kadi & Marufuzzaman, 2013*). As a time-frequency representation of a signal, WDec has some advantages over other conventional techniques as optimal resolution in both the time and frequency domains and lack of the requirement of stationarity of the signal (*Akar et al., 2015*; *Quiroga et al., 2001*). The PSDs of the extracted alpha rhythms from a simulated EEG signal by the FIR with order 60 and WDec methods are shown in Figs. 5A and 5B, respectively. The PSDs of the extracted alpha rhythm by the FIR and WDec methods had a higher magnitude, and there were more component mixing than that by the CiSSA method. The errors of the extracted alpha rhythm by the FIR and WDec method were $\varepsilon = 0.51\ \mu V^2/\text{Hz}$ and $\varepsilon = 0.57\ \mu V^2/\text{Hz}$, respectively, which were higher than that obtained by the CiSSA method. This is because other rhythms and artifacts (electrooculogram (EOG), electromyography (EMG), baseline drift and stochastic noise) also have components in the alpha band (8–13 Hz). The FIR filter and the WDec methods are not able to separate the component mixing overlapping in frequency space. However, the CiSSA method can suppress a part of noise with overlapping frequencies.

**Table 2 The average errors of the extracted alpha rhythm by the CiSSA, basic SSA, FIR and WDec methods.**

| Method | $\bar{\varepsilon}(\mu V^2/\text{Hz})$ |
|---|---|
| CiSSA ($L = 80$) | $0.29 \pm 0.08$ |
| basic SSA ($L = 80$) | $0.40 \pm 0.18$ |
| FIR | $0.49 \pm 0.23$ |
| WDec | $0.58 \pm 0.21$ |

**Table 3 The average errors of the extracted alpha rhythm by the CiSSA method with different embedding dimensions.**

| $L$ | $\bar{\varepsilon}(\mu V^2/\text{Hz})$ | $L$ | $\bar{\varepsilon}(\mu V^2/\text{Hz})$ | $L$ | $\bar{\varepsilon}(\mu V^2/\text{Hz})$ | $L$ | $\bar{\varepsilon}(\mu V^2/\text{Hz})$ |
|---|---|---|---|---|---|---|---|
| 20 | $0.82 \pm 0.15$ | 70 | $0.56 \pm 0.14$ | 120 | $0.50 \pm 0.12$ | 170 | $0.50 \pm 0.11$ |
| 30 | $0.89 \pm 0.40$ | 80 | $0.29 \pm 0.08$ | 130 | $0.33 \pm 0.09$ | 180 | $0.38 \pm 0.10$ |
| 40 | $0.39 \pm 0.09$ | 90 | $0.44 \pm 0.12$ | 140 | $0.45 \pm 0.10$ | 190 | $0.45 \pm 0.10$ |
| 50 | $0.76 \pm 0.19$ | 100 | $0.59 \pm 0.14$ | 150 | $0.54 \pm 0.12$ | 200 | $0.54 \pm 0.11$ |
| 60 | $0.33 \pm 0.09$ | 110 | $0.38 \pm 0.10$ | 160 | $0.40 \pm 0.10$ | | |

In order to increase the robustness and validity of alpha rhythm extraction, 1,000 simulated EEG signals have been performed and the average error of the PSD between the simulated alpha rhythm and the extracted alpha rhythm was calculated as

$$\bar{\varepsilon} = \varepsilon_{\text{mean}} \pm \varepsilon_{\text{std}} \qquad (14)$$

where $\varepsilon_{\text{mean}}$ is mean of errors ($\varepsilon_{ave}$) in 1,000 simulations, $\varepsilon_{\text{std}}$ is the standard deviation (STD) of errors ($\varepsilon_{ave}$) in 1,000 simulations. The average errors of the extracted alpha rhythm by the CiSSA, basic SSA, FIR and WDec methods are shown in Table 2. The mean and STD of the errors of the PSD between the simulated alpha rhythm and the extracted alpha rhythm by the CiSSA method ($L = 80$) were $\varepsilon = 0.29$ $\mu V^2/\text{Hz}$ and $\varepsilon = 0.08$ $\mu V^2/\text{Hz}$, respectively, which were much lower than those obtained by the basic SSA, FIR and WDec methods.

We conclude that the RCs of simulated EEG signals processed by the CiSSA method fall on the theoretical frequencies limited to the selected bandwidth ranges. The alpha rhythm can be extracted automatically based on the frequency feature. The alpha rhythm extraction by the CiSSA method performed better than that of the basic SSA, FIR, and WDec methods. Therefore, the processing results of the simulated EEG verify the validity of the CiSSA method's performance in alpha rhythm extraction. Furthermore, the error of the extracted alpha rhythm by the CiSSA method varied with the embedding dimension. The calculated average errors of the extracted alpha rhythm from 1,000 simulated EEG signals by the CiSSA method with different embedding dimensions are shown in Table 3. The average error attained a minimum value at $L = 80$. Therefore, the embedding dimension of the CiSSA method for alpha extraction was set to $L = 80$.

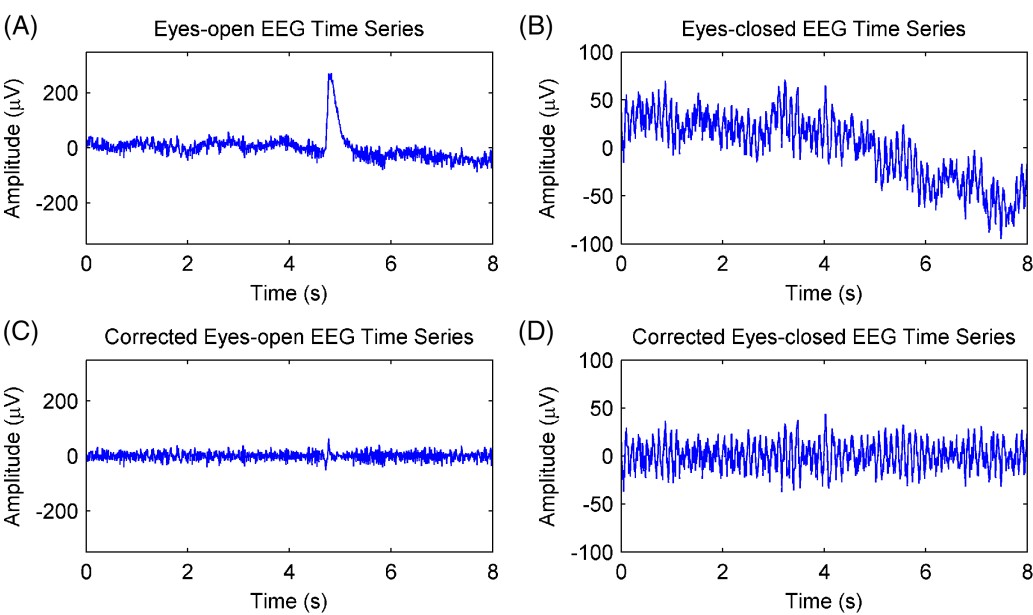

**Figure 6 Artifacts removal of EEG signals.** (A) A raw EEG epoch of subject 17# in eyes-open condition. (B) A raw EEG epoch of subject 17# in eyes-closed condition. (C) The corrected eyes-open EEG signal after artifact removal. (D) The corrected eyes-closed EEG signal after artifact removal.

# EXPERIMENTAL RESULTS AND DISCUSSION

## Results and discussion of database EEG signals

The database EEG signals reported in literature (*Trujillo, Stanfield & Vela, 2017*) were used. A total of 22 subjects (11 male and 11 female) aged from 18–26 participated in the experiment. The experiments were carried out with the subjects sitting quietly on a comfortable chair in a darkened room. Every subject underwent 8 min of eyes-open and eyes-closed resting state, with 4 min eyes open and 4 min eyes closed interleaved in 1-min intervals. The EEG signals of one subject (subject #6) were removed because of a technical recording error. A total of 72 channels of EEG signals were recorded by a BioSemi Active II amplifier system. The EEG signals were downsampled to 256 Hz.

Channel Fpz was selected for EEG analysis. The EEG data of the Fpz channel were divided into 8-s (2,048 samples) epochs with 50% overlap, thus producing 91 epochs of eyes-open and eyes-closed conditions for each subject. This was done because artifacts, including those resulting from electrooculogram, electromyography, baseline drift, and stochastic noise, interfere with the rhythm extraction. The adaptive SSA method (*Hai et al., 2017*) was used to remove the artifacts, and the results are shown in Fig. 6. Figures 6A and 6B show EEG epochs of the eyes-open and eyes-closed conditions of subject #17, respectively. Figures 6C and 6D show the corrected EEG signals after artifact removal. The electrooculogram artifacts were removed from the EEG signals of the eyes-open condition, and the spontaneous EEG signals were preserved in both conditions.

The EEG signals after artifact removal were processed by the CiSSA method with the embedding dimension set to $L = 80$. The first six RCs of the EEG signals in the eyes-open

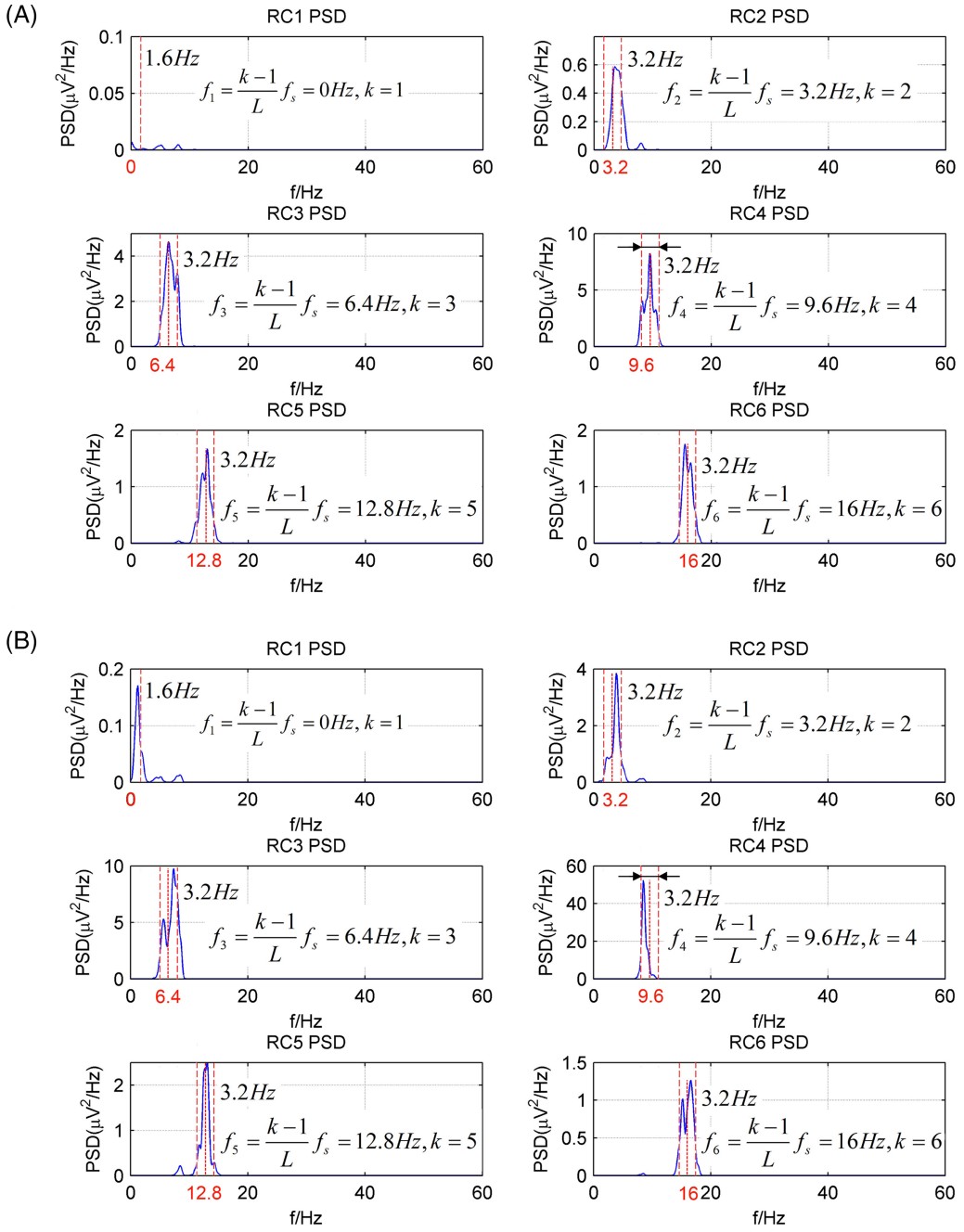

**Figure 7 The PSD of first six RCs of real EEG signals processed by the CiSSA method ($L = 80$).**
(A) Eyes-open condition and (B) eyes-closed condition.

and eyes-closed conditions are shown in Figs. 7A and 7B, respectively. Each RC falls on the theoretical frequency derived by Eq. (5), and the bandwidth of the RCs is limited to $f_s/L = 3.2$ Hz, which is agreement with the simulation results. Figures 7A and 7B show that RC4 and RC5 represent the alpha rhythm in both the eyes-open and eyes-closed conditions, respectively. Thus, the alpha rhythms of the EEG signals can be extracted automatically as the sum of RC4 and RC5, which is agreement with Eq. (8). Figures 8A

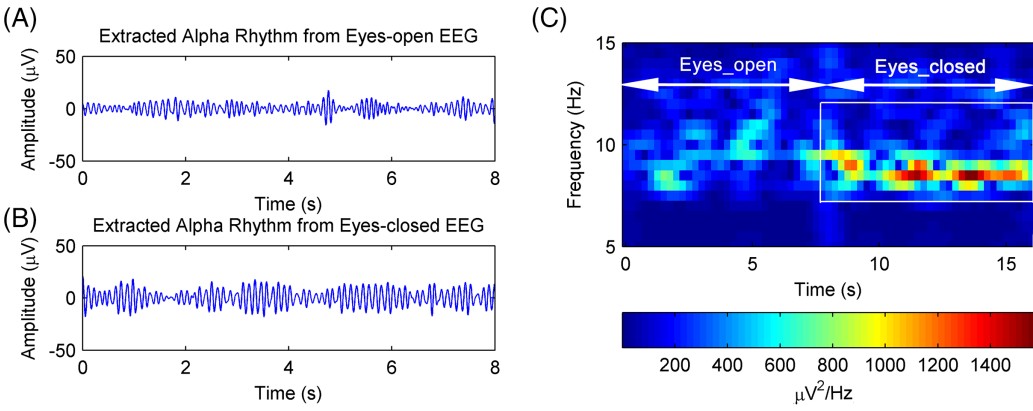

**Figure 8 The extracted alpha rhythms of real EEG signals.** (A) Eyes-open and (B) eyes-closed condition and the (C) the spectrogram of alpha rhythms.

and 8B show the extracted alpha rhythms of the EEG signals in the eyes-open and eyes-closed conditions, respectively. The amplitude of the alpha rhythm in the eyes-open condition was lower than that in the eyes-closed condition. This was consistent with the results of previous studies, in which the alpha rhythm in the resting state in the eyes-open condition with visual stimulation was much weaker than that in the eyes-closed condition (*Barry et al., 2007*). Figure 8C illustrates the spectrogram of alpha rhythms in the eyes-open and eyes-closed conditions, which is the square of the rhythm's amplitude as a function of time and frequency. It illustrates a significant difference between the eyes-open and eyes-closed states.

The performance of alpha rhythm extraction by the CiSSA method was compared with that of three other methods: the basic SSA method, the WDec method, and the FIR method. The alpha rhythms under the eyes-open and eyes-closed conditions were extracted using the CiSSA, basic SSA, WDec, and FIR methods. The PSD of the extracted alpha rhythms by the four methods under the eyes-closed and eyes-open conditions are shown in Fig. 9. Figures 9A and 9D show that the extracted alpha rhythms using the CiSSA method were within the alpha band (8–13 Hz) under both the eyes-open and eyes-closed conditions. In addition, the power of the extracted alpha rhythm under the eyes-open condition was lower than that under the eyes-closed condition. Therefore, the alpha rhythm extracted using the CiSSA method could represent the real EEG alpha rhythm. However, the alpha rhythm extracted by the basic SSA method under both the eyes-open and eyes-closed conditions contained frequency components less than 8 Hz, which were outside the alpha band. Similarly, the alpha rhythms extracted using the WDec method under eyes-closed condition contained components outside the alpha band (component mixing), as shown in Fig. 9E. Figures 9B, 9C and 9F show the alpha rhythms extracted using the WDec method under eyes-open condition, using the FIR method under eyes-open and eyes-closed conditions, respectively. The extracted alpha rhythms fell into the alpha band, and that the power of the extracted alpha rhythm was stronger than that extracted using the CiSSA method. This is in agreement with the simulation results shown in Fig. 5 because the WDec and FIR methods are unable to remove artifacts

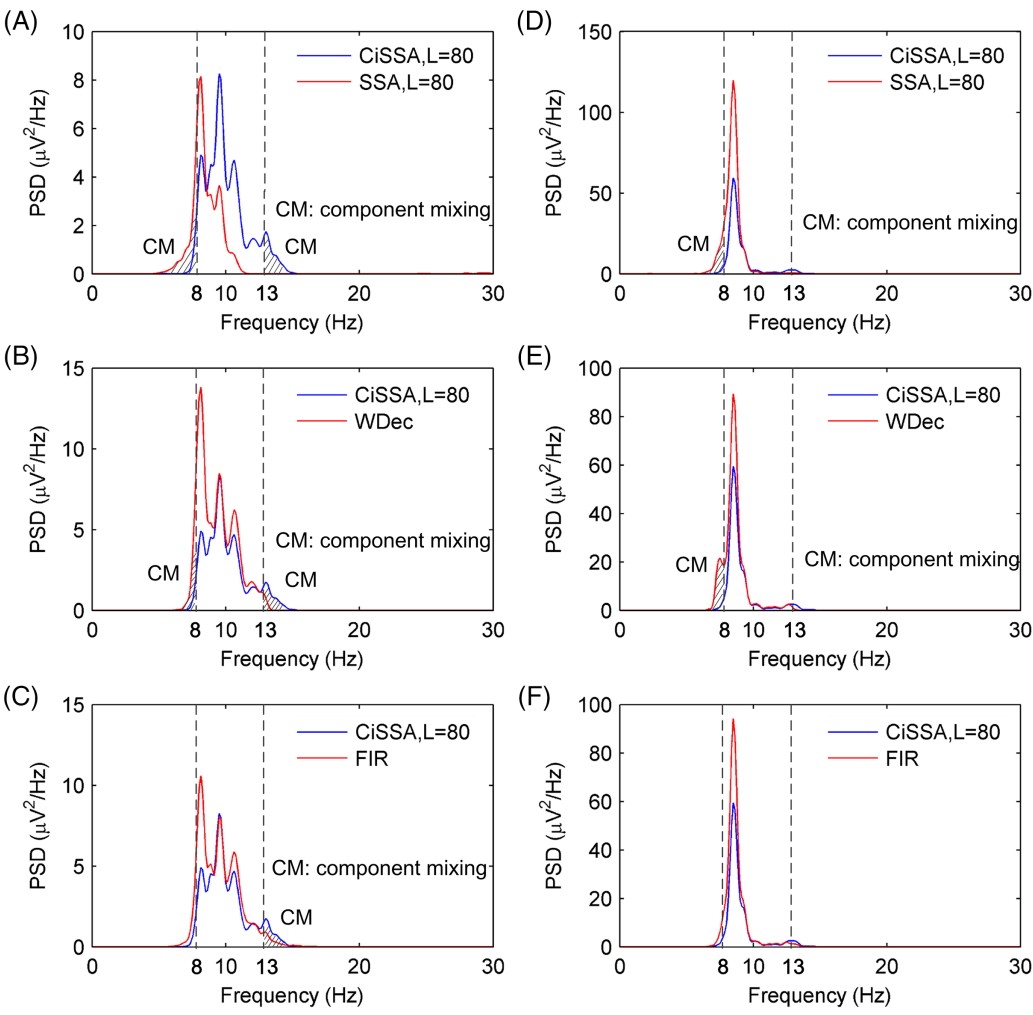

**Figure 9 The PSD of the extracted alpha rhythms using the CiSSA, basic SSA, WDec and FIR methods.** The PSD of the extracted alpha rhythms using the CiSSA and basic SSA method under (A) eyes-open and (D) eyes-closed conditions; the PSD of the extracted alpha rhythms using the CiSSA and WDec method under (B) eyes-open and (E) eyes-closed conditions; the PSD of the extracted alpha rhythms using the CiSSA and FIR method under (C) eyes-open and (F) eyes-closed conditions.

and noise from the alpha rhythm with an overlapping frequency spectrum. Therefore, the CiSSA method performed better than the basic SSA, WDec, and FIR methods at alpha rhythm extraction.

To further verify the CiSSA method's performance, the extracted alpha rhythms were used to distinguish between the eyes-open and eyes-closed states, and the classification results produced by the CiSSA method were compared with those by the basic SSA, FIR, and WDec methods. In this study, the power $\left(P = \sum_{i=1}^{N} V_i^2/N\right)$ and the mean of the absolute value $\left(\bar{V} = \sum_{i=1}^{N} |V_i|/N\right)$ were selected as the features of the alpha rhythm (*Mohammadi et al., 2015*), where $V_i$ represents the amplitude of the extracted alpha rhythm, and $N$ represents the number of samples. Figure 10A shows the values of the power and the mean of the absolute value of the extracted alpha rhythm by the CiSSA

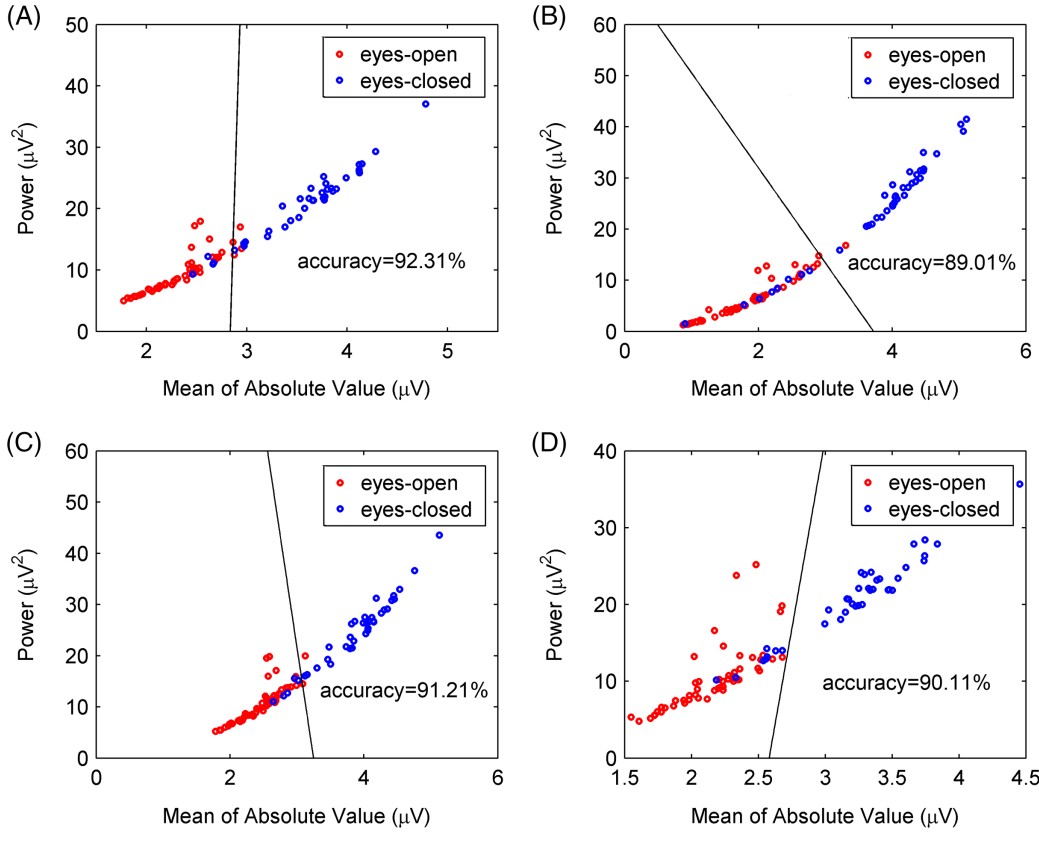

**Figure 10 Classification results for subject 17# between eyes-open and eyes-closed states.** (A) The CiSSA method ($L = 80$), (B) the basic SSA method ($L = 80$), (C) the FIR method and (D) the WDec methods.               

method for subject #17. The power value and the mean of the absolute value under the eyes-open condition were lower than those under the eyes-closed condition. Then, the support vector machine method was used to classify the features under the eyes-open and eyes-closed conditions. The classification accuracy was 92.31%. Figures 10B–10D show the power values and mean absolute values of the extracted alpha rhythms by the basic SSA, FIR, and WDec methods, respectively. Similar to the results obtained by the CiSSA method, the power and mean of the absolute value of the extracted alpha signal under the eyes-open condition were generally lower than those under the eyes-closed condition. The classification accuracy of feature extraction by the basic SSA, FIR, and WDec methods was 89.01%, 91.21%, and 90.11%, respectively, and these values were lower than the accuracy obtained by the CiSSA method.

  We calculated the power values and means of the absolute value of the extracted alpha rhythms of all 21 subjects, and the classification results are shown in Table 4. The classification accuracy varied greatly between different subjects because of individual differences in EEG signals. The mean and standard deviation of the classification accuracy was calculated for all subjects to compare classification performance between the CiSSA, basic SSA, FIR, and WDec methods. The mean value of the classification accuracy for all subjects by the CiSSA method was 92.36%, which was higher than those obtained by

**Table 4 Classification accuracies for all subjects by the CiSSA, basic SSA, FIR and WDec methods.**

| Subject # | CiSSA ($L = 80$) (%) | Basic SSA ($L = 80$) (%) | FIR (%) | WDec (%) |
|---|---|---|---|---|
| Subject 1 | 80.22 | 79.12 | 79.12 | 79.12 |
| Subject 2 | 95.60 | 67.03 | 94.51 | 83.52 |
| Subject 3 | 96.70 | 58.24 | 97.80 | 95.60 |
| Subject 4 | 95.60 | 94.51 | 92.31 | 94.51 |
| Subject 5 | 98.90 | 95.60 | 98.90 | 98.90 |
| Subject 7 | 96.70 | 96.70 | 93.41 | 94.51 |
| Subject 8 | 100 | 100 | 100 | 100 |
| Subject 9 | 81.32 | 71.43 | 78.02 | 74.73 |
| Subject 10 | 100 | 100.00 | 98.90 | 100 |
| Subject 11 | 96.70 | 95.60 | 95.60 | 95.60 |
| Subject 12 | 76.92 | 70.33 | 74.73 | 72.53 |
| Subject 13 | 100 | 93.41 | 98.90 | 100 |
| Subject 14 | 87.91 | 87.91 | 87.91 | 87.91 |
| Subject 15 | 95.60 | 93.41 | 93.41 | 96.70 |
| Subject 16 | 86.81 | 86.81 | 86.81 | 86.81 |
| Subject 17 | 92.31 | 89.01 | 91.21 | 90.11 |
| Subject 18 | 95.60 | 91.21 | 98.90 | 97.80 |
| Subject 19 | 83.52 | 75.82 | 84.62 | 81.32 |
| Subject 20 | 98.90 | 100.00 | 98.90 | 98.90 |
| Subject 21 | 94.51 | 94.51 | 93.41 | 94.51 |
| Subject 22 | 85.71 | 83.52 | 85.71 | 85.71 |
| Average | 92.36 | 86.87 | 91.58 | 90.89 |
| STD | 7.05 | 11.78 | 7.42 | 8.39 |

the basic SSA (86.87%), FIR (91.58%), and WDec (90.89%) methods. The standard deviation of the classification accuracy across all subjects by the CiSSA method was 7.05%, which was lower than those obtained by the basic SSA (11.78%), FIR (7.42%), and WDec (8.39%) methods. Therefore, the CiSSA method's classification performance was better and more robust than that by the basic SSA, FIR, and WDec methods.

## Results and discussion of experimental EEG signals

Additional experimental EEG signals were recorded and used to further verify the validity of the CiSSA method. The Institutional Review Board of Tsinghua University granted ethical approval (20170010) to carry out the experiment within its facilities. Ten subjects (8 male and 2 female) aged 20–29 participated in the experiments and the written informed consent was obtained from the subjects. The experiments were carried out with the subjects sitting on a comfortable chair in a room with normal lightness. The experimental EEG signals were recorded using the MP160 data acquisition and analysis system (BIOPAC Systems, Inc., Goleta, CA, USA) with the sampling rate of 200 Hz. The active Ag/AgCl electrode flushed with conductive gel was used as the recording electrode and attached to the frontal region of the subject's scalp. Another two electrodes, which served

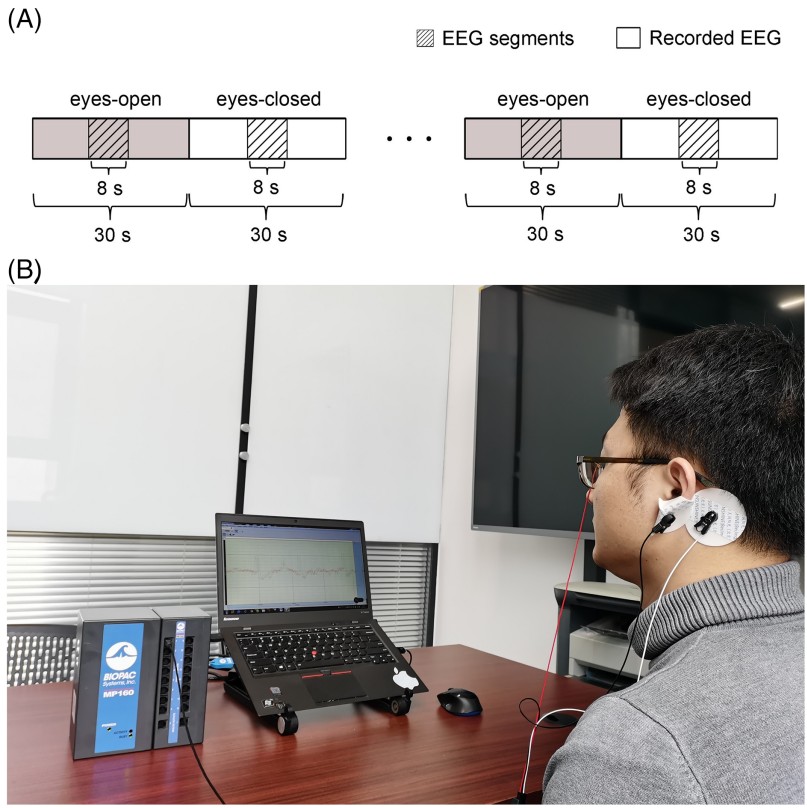

**Figure 11 Set up of the experiment.** (A) Schematic of the recorded EEG data and (B) the photograph of the experiment. A total of 114 times of alternating periods of 30 s eyes open followed by 30 s eyes closed. The desired EEG segments were cut off in the middle of every period of the eyes-open and eyes-closed states. Each segment last for 8 s.

as ground and reference, were attached to the earlobe and mastoid, respectively, as shown in Fig. 11B. Every subject underwent 30 s of eyes-open resting state, followed by 30 s of eyes-closed resting state, and repeated this procedure 57 times. Segments lasting 8 s were extracted from the middle of each period, as shown in Fig. 11A. Consequently, 57 segments each of the eyes-open and eyes-closed states for each subject were obtained. Artifacts were removed from each EEG segment using the adaptive SSA method (*Hai et al., 2017*).

The experimental EEG signals were processed by the CiSSA method, and the alpha rhythms recorded under the eyes-open and eyes-closed conditions were extracted. The power and mean absolute values of the extracted alpha rhythms were calculated as features for classification using the support vector machine method. The classification accuracy by the CiSSA method for subject 1# was 91.23%, which was higher than that obtained by the basic SSA (87.72%), FIR (89.47%), and WDec (88.60%) methods, as shown in Fig 12. The classification results of all 10 subjects are shown in Table 5. The mean value of the classification accuracy for all subjects by the CiSSA method was 92.11%, which was higher than those obtained by the basic SSA (87.46%), FIR (90.17%), and WDec (88.77%) methods. The standard deviation of the classification accuracy across all subjects by the CiSSA method was 4.74%, which was lower than those obtained by the basic SSA (5.86%),

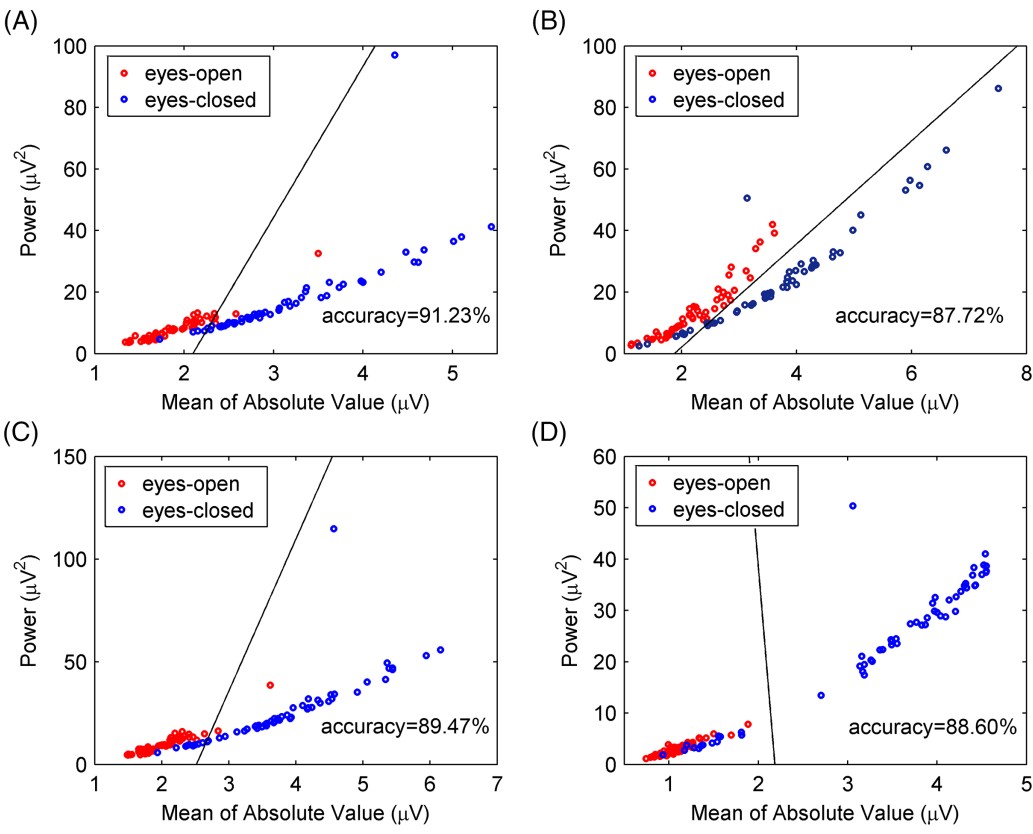

**Figure 12 Classification results for experimental EEG signals between eyes-open and eyes-closed states.** (A) The CiSSA method ($L$ = 80), (B) the basic SSA method ($L$ = 80), (C) the FIR method and (D) the WDec methods.

**Table 5 Classification accuracies of experimental EEG signals for all subjects by the CiSSA, basic SSA, FIR and WDec methods.**

| Subject # | CiSSA ($L$ = 80) (%) | Basic SSA ($L$ = 80) (%) | FIR (%) | WDec (%) |
|---|---|---|---|---|
| Subject 1 | 91.23 | 87.72 | 89.47 | 88.60 |
| Subject 2 | 92.98 | 93.86 | 95.61 | 95.61 |
| Subject 3 | 95.61 | 91.23 | 95.61 | 95.61 |
| Subject 4 | 93.86 | 80.7 | 88.6 | 84.21 |
| Subject 5 | 92.11 | 84.21 | 84.21 | 96.49 |
| Subject 6 | 86.84 | 86.84 | 84.21 | 89.47 |
| Subject 7 | 96.49 | 85.09 | 93.86 | 94.74 |
| Subject 8 | 97.37 | 98.25 | 99.12 | 99.12 |
| Subject 9 | 80.7 | 77.19 | 79.82 | 66.67 |
| Subject 10 | 93.86 | 89.47 | 91.23 | 77.19 |
| Average | 92.11 | 87.46 | 90.17 | 88.77 |
| STD | 4.74 | 5.86 | 5.79 | 9.69 |

FIR (5.79%), and WDec (9.69%) methods. We therefore concluded that the CiSSA method's classification performance was better than that by the basic SSA, FIR, and WDec methods.

## CONCLUSIONS

In this paper, a flexible and accurate method based on CiSSA was proposed for alpha rhythm extraction from EEG signals. By decomposing the EEG signals into a set of orthogonal reconstructed components (RCs) at specific bandwidths of frequencies, the alpha rhythm can be extracted flexibly and accurately from EEG signals. The proposed method performed well on both simulated EEG data generated from the MPA EEG model and experimental EEG data, as well as the EEG data obtained from a public database. Features of the alpha rhythms extracted from experimental EEG signals were calculated to distinguish between the eyes-open and eyes-closed states. The CiSSA-based method showed higher classification accuracy and robustness than that of the basic SSA, FIR and WDec methods.

### Funding
This work was supported by the National Key Research and Development Program of China (No. 2018YFB2003201). The funders had no role in study design, data collection and analysis, decision to publish, or preparation of the manuscript.

### Grant Disclosures
The following grant information was disclosed by the authors:
National Key Research and Development Program of China: 2018YFB2003201.

### Competing Interests
The authors declare that they have no competing interests.

### Author Contributions
- Hai Hu conceived and designed the experiments, performed the experiments, analyzed the data, prepared figures and/or tables, authored or reviewed drafts of the paper, and approved the final draft.
- Zihang Pu performed the experiments, authored or reviewed drafts of the paper, and approved the final draft.
- Peng Wang conceived and designed the experiments, prepared figures and/or tables, authored or reviewed drafts of the paper, and approved the final draft.

### Human Ethics
The following information was supplied relating to ethical approvals (*i.e.*, approving body and any reference numbers):
The Institutional Review Board of Tsinghua University granted Ethical approval to carry out the study within its facilities.
## Data Availability

The MATLAB codes and the raw experimental EEG data are available in the Supplemental Files.

The data is also available at Trujillo, Logan, 2020, "Raw EEG Data", https://doi.org/10.18738/T8/SS2NHB, Texas Data Repository, V1.

## Supplemental Information

Supplemental information for this article can be found online at http://dx.doi.org/10.7717/peerj.13096#supplemental-information.

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
