# Peer review of "A flexible and accurate method for electroencephalography rhythms extraction based on circulant singular spectrum analysis"

_PeerJ, doi:10.7717/peerj.13096_

## Round 0.1 · original submission · Major Revisions

I warmly suggest you take care of all the issues raised by the Reviewers.

Reviewer 1 ·

Basic reporting

Comments included in the attached file.

Experimental design

Comments included in the attached file.

Validity of the findings

Comments included in the attached file.

Additional comments

Comments included in the attached file.

Annotated reviews are not available for download in order to protect the identity of reviewers who chose to remain anonymous.

·

Basic reporting

In this study, authors have applied circulant singular spectrum analysis method to decompose an electroencephalography (EEG) signal in to a set of orthogonal reconstructed components, then the reconstructed components are then grouped flexibly to extract the desired EEG rhythms based on the known frequencies.

Simulated EEG data and real EEG data have been used to test the performance of this addressed method, which is better than basic singular spectrum analysis method, the wavelet decomposition method, and the infinite impulse response filtering method.

Experimental design

The simulated EEG and real EEG data with the closed and open eys states have been tested.

Validity of the findings

The comparison with other methods has been used to valid the performance of this proposed method.

Additional comments

No comments.

Reviewer 3 ·

Basic reporting

This is an interesting manuscript on using time-delay embedding to isolate spectral components in a signal. This is not the first application of decomposing time-delay embedded matrices for neural signal processing, but it does provide some new information that could be useful. Below are my comments.

The authors repeatedly claim that the extracted rhythms do not include mixing from other frequencies or artifacts. I don't really understand the significance of the first point -- band-pass filtering also excludes other frequencies (within the range of the passband frequencies and filter transition zones, etc). Furthermore, how does the CSSA method exclude artifacts? Broadband artifacts still have energy in, e.g., the alpha band, and some artifacts such a blinks are concentrated in low frequencies such as delta/theta. All of the methods I'm aware of that isolate narrowband activity while suppressing artifacts are spatial multivariate methods that leverage the linearly dissociable spatial distribution of artifacts vs. signal. That does not seem to be the case here.

Please add some information about how to set L. Setting it arbitrarily to 40 seems like a bad idea, because it depends entirely on the data sampling rate and the desired bandwidth. I can imagine two approaches: (1) Use eq 9 to get L = Fs/Fb. So then users can specify their desired bandwidth to obtain L. (2) Define L based on the number of cycles at the desired center frequency. For example, the center of alpha is 10 Hz, and having, e.g., two cycles at 10 Hz would give a decent spectral estimation (if the SNR is sufficient), so then L should be at least 200 for a sampling rate of 1 kHz. Thinking about it this way, L=40 with Fs=1000 would lead to only .4 of one cycle of alpha, which isn't enough to get any reliable results.

I don't think that the comparison shown in Figure 5 is fair. The bandwidth of CSSA was narrower than the bandwidths of the IIR and wavelet filters. So it's trivial that the filters will pass through energy at a wider frequency range, which here leads to mixing. A narrower filter (and probably a better one, e.g., FIR) would give (I guess) comparable performance to CSSA. By the same token, if the authors had simulated the data to have a wider alpha spectral peak but used the same L parameter, then IIR would have given more accurate results because the CSSA bandwidth would have been too narrow.
And I don't understand what's happening with the wavelet in panel B. The wavelet appears to pass through frequencies from 4-10 Hz, so obviously it will miss the alpha. It's an inappropriate filter, so what's the point of showing it here?

Same comment for Figure 9: The wavelet is passing through a strange collection of frequencies. This is not the appropriate way to do wavelet convolution in electrophysiology data; a useful wavelet would have narrowband characteristics, e.g., a Morlet or Gabor wavelet.

I have to admit, I was more enthusiastic about the study from the intro/methods than when seeing the results. This results appear to show that CSSA is effectively a narrowband filter. I don't think this should preclude publication, but I think the authors need to include a more detailed description of the implications, possible uses, and advantages of CSSA over a carefully designed narrowband filter, which is more commonly done in neuroscience, and which is also computationally more efficient.

Experimental design

.

Validity of the findings

.

---

## Round 0.2 · accepted · Accept

Reviewers consider that the present version has substantially improved and that you answered all the questions they posed.

Reviewer 1 ·

Basic reporting

Comments included in the attached file.

Experimental design

Comments included in the attached file.

Validity of the findings

Comments included in the attached file.

Additional comments

Comments included in the attached file.

Annotated reviews are not available for download in order to protect the identity of reviewers who chose to remain anonymous.

Reviewer 3 ·

Basic reporting

No comment

Experimental design

No comment

Validity of the findings

No comment

Additional comments

No comment